# Tight Junction Protein 1a regulates pigment cell organisation during zebrafish colour patterning

Andrey Fadeev, Jana Krauss[†], Hans Georg Frohnhöfer, Uwe Irion, Christiane Nüsslein-Volhard*

Max Planck Institute for Developmental Biology, Tübingen, Germany

**Abstract** Zebrafish display a prominent pattern of alternating dark and light stripes generated by the precise positioning of pigment cells in the skin. This arrangement is the result of coordinated cell movements, cell shape changes, and the organisation of pigment cells during metamorphosis. Iridophores play a crucial part in this process by switching between the dense form of the light stripes and the loose form of the dark stripes. Adult *schachbrett* (*sbr*) mutants exhibit delayed changes in iridophore shape and organisation caused by truncations in Tight Junction Protein 1a (ZO-1a). In *sbr* mutants, the dark stripes are interrupted by dense iridophores invading as coherent sheets. Immuno-labelling and chimeric analyses indicate that Tjp1a is expressed in dense iridophores but down-regulated in the loose form. Tjp1a is a novel regulator of cell shape changes during colour pattern formation and the first cytoplasmic protein implicated in this process.

*For correspondence: christiane.
nuesslein-volhard@tuebingen.
mpg.de

Present address: [†]Auengrund 7,
Schönau-Berzdorf, Germany

Competing interests: The
authors declare that no
competing interests exist.

Reviewing editor: Marianne E
Bronner, California Institute of
Technology, United States

## Introduction

One of the most fascinating features of vertebrates is their display of remarkable colour patterns in skin, fur, or plumage, frequently varying strikingly between closely related species. Teleost fish exhibit a particularly high diversity of patterns formed by several types of pigment cells distributed in a multilayered arrangement in the hypodermis (*Singh and Nüsslein-Volhard, 2015*). Adult zebrafish display a conspicuous pattern of alternating dark and light stripes; remarkably different from a relatively simple larval pattern, which is generated directly from neural crest cells migrating during embryogenesis (*Kelsh et al., 1996*). The adult pattern is formed from neural crest-derived progenitors during metamorphosis (3–6 weeks of development). Metamorphic iridophores (silvery cells containing reflective guanine platelets) and melanophores (dark cells containing the black pigment melanin) arise from neural crest-derived stem cells associated with the peripheral nervous system, whereas metamorphic xanthophores (yellow–orange cells containing pteridine based pigments) originate from proliferating larval xanthophores (*Budi et al., 2011*; *Dooley et al., 2013*; *Mahalwar et al., 2014*; *McMenamin et al., 2014*; *Singh et al., 2014*). Several adult viable zebrafish mutants displaying abnormal adult pigment patterns have been described (*Haffter et al., 1996*; *Kelsh et al., 1996*; *Lister et al., 1999*). One class of genes primarily affects the formation of one of the three cell types. For example *nacre/mitfa* mutants lack melanophores, *pfeffer/csf1ra/fms* mutants lack xanthophores, and in *shady/ltk* iridophores are compromised (*Lister et al., 1999*; *Parichy et al., 2000*; *Lopes et al., 2008*). Genetic analyses and regeneration studies revealed that interactions between all three cell types are necessary for proper stripe formation in the trunk of the fish (*Maderspacher and Nüsslein-Volhard, 2003*; *Nakamasu et al., 2009*; *Frohnhöfer et al., 2013*; *Patterson and Parichy, 2013*).

Long-term in vivo imaging has shown that stripe formation involves intricate cell shape and density changes of metamorphic pigment cells (*Mahalwar et al., 2014*; *Singh et al., 2014*). Iridophores take a lead in stripe formation: they appear along the horizontal myoseptum, proliferate and spread as

**eLife digest** The striking horizontal striped pattern of the zebrafish makes it a decorative addition to many home aquariums. The stripes are a result of three different pigment cells interacting with each other, and first begin to emerge when the animal is two to three weeks old. At that time, iridescent cells called iridophores begin to multiply and spread in the skin. In the light-coloured stripes, the iridophores are compact and 'dense'; in the dark stripes the cells change into a 'loose' shape and organisation. Black-pigmented cells fill in the dark stripes, and a third cell type with a yellow hue condenses over the light stripes. How the three types of cell work together to make the striped pattern is not fully understood.

Fadeev et al. examined a zebrafish variant with a genetic mutation that disrupts the function of a protein called Tight Junction Protein 1a (or Tjp1a)—a fish variant of a mammalian protein called ZO-1. This protein helps cells to interact with each other. The mutant fish appear spotted rather than striped, because light regions containing sheets of the dense iridophores interrupt the dark stripes.

Experiments using fluorescent markers showed that Tjp1a is produced in much lower amounts in the loose iridophores in the dark stripes than in the dense iridophores of the light stripes. This led Fadeev et al. to suggest that the transition from the dense to the loose shape is dependent on the presence of Tjp1a in the cell.

Tjp1a is likely to regulate how colour patterns form by controlling how iridophores interact with other types of pigment cell. The Tjp1a mutant fish provides the first glimpse into the machinery inside cells that underlies colour pattern formation, and will help to identify other components and cues responsible for cell interactions.

a dense sheet in the skin to form the first light stripe. At the margins of this first light stripe, the dense iridophores undergo a transition into a loose form and spread over the dark stripe region. Past the presumptive dark stripe, they change into the dense form again and aggregate into sheets forming new light stripes (*Singh et al., 2014*). The first two dark stripes form dorsally and ventrally of the first light stripe by melanoblasts migrating along spinal nerves into the skin in the presumptive dark stripe region. They initially appear as stellate cells with the pigment located in the centre of the cells but later expand into the stationary rounded form (*Dooley et al., 2013*; *Singh et al., 2014*). Metamorphic xanthophores originate from larval xanthophores, they compact over the dense iridophores of the light stripe and change into a pale stellate shape above the loose iridophores and melanophores of the dark stripe (*Mahalwar et al., 2014*). A different type of iridophores—L-iridophores—underlie the melanophores of the dark stripe. L-iridophores appear only after the first two dark stripes are formed and do not participate in laying out the pattern (*Frohnhöfer et al., 2013*; *Hirata et al., 2003*, *2005*). Interestingly, iridophore-deficient mutants are not affected in the stripe pattern of the fins, suggesting differences in the mechanisms involved in patterning of the trunk and fins (*Frohnhöfer et al., 2013*).

Mutants in which all three chromatophore types develop, but stripe formation is impaired, are of particular interest, as they can provide insights in the molecular mechanisms of cell–cell interactions underlying stripe formation. Several mutants have been described in which dark stripes are broken into spots. *leopard/Cx 41.9*, *luchs/Cx39.4* encode components of gap junctions involved in cell–cell communications (*Maderspacher and Nüsslein-Volhard, 2003*; *Watanabe et al., 2006*; *Irion et al., 2014*). In the absence of *leo* or *luc*, iridophores fail to change to the loose form and suppress melanophores. *leo* and *luc* presumably form heteromeric gap junctions among and between melanophores and xanthophores, instructing iridophores to change shape in a spatially controlled manner (*Irion et al., 2014*).

In this study, we present the mutant *schachbrett* (*sbr*) (German for checkerboard) that exhibits interruptions in dark stripes by light stripe regions. *sbr* encodes Tight Junction Protein 1a (Tjp1a/ZO-1). Immunostaining revealed that Tjp1a is expressed in dense iridophores but neither in loose iridophores nor any other pigment cell type. Analysis of double mutants and chimeras shows that *sbr* is cell-autonomously required in iridophores. During metamorphosis, dense iridophores invade the dark stripe regions and temporarily suppress the expansion of melanophores, suggesting that Tjp1a is required to regulate the transition of dense iridophores into the loose shape and their organisation.

# Results

## *schachbrett* encodes Tight Junction Protein 1a

Adult *sbr* fish display an unchanged arrangement and approximately normal width of stripes, however, the dark stripes are interrupted and undulating (**Figure 1A**). The allele *sbr^tnh009b^* was isolated during a screen for ENU-induced recessive, homozygous viable mutants affecting adult pattern formation. The mutation was mapped to the region 29.6–32.5 Mb of chromosome 7 (Ensembl Zebrafish release 72) (**Figure 1B**). Using a candidate approach, we sequenced *tjp1a* cDNA of *sbr^tnh009b^* and detected a nonsense mutation leading to Y1143Stop change in the C-terminal part of the protein (**Figure 1C**). To confirm the suggestion that this mutation is causative for the *sbr* phenotype, we performed a screen for additional alleles. ENU-mutagenized Tü males were crossed to *sbr^tnh009b^* females; the progeny was raised to the adulthood and screened for the *sbr* phenotype. Four new

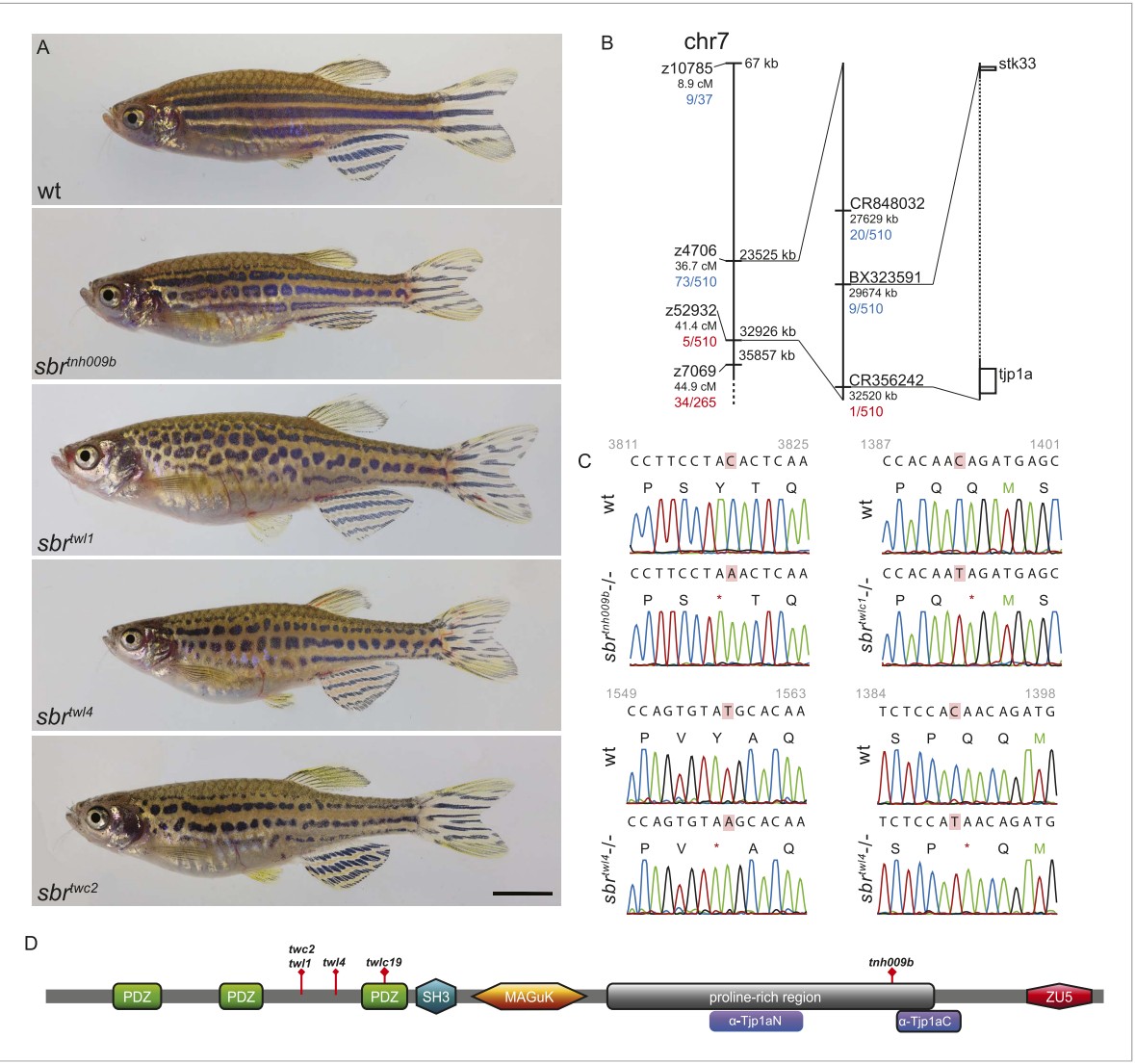

**Figure 1**. *schachbrett* encodes Tjp1a. (**A**) All alleles of *sbr* exhibit interrupted, undulating dark stripes of normal arrangement and width when compared to wild type, but no other obvious defects. Scale bar: 5 mm. (**B**) Scheme of meiotic mapping of *sbr*. Marked are z-markers and contigs on which SNPs were found with their genomic and genetic (where applicable) coordinates. The numbers of recombinants among all fish tested are given in red and blue. The right-most bar shows genes on the ends of the final mapped region. The dotted region is not to scale and contains multiple genes. (**C**) DNA sequence traces for four alleles of *sbr*. Red rectangles mark the mutated residues. Red asterisks stand for stop codons. (**D**) Scheme of Tjp1a protein. Purple rounded squares indicate regions corresponding to polypeptides used for antibody generation. Red diamonds show the positions of stop codons in the mutants.

alleles not complementing the original allele were isolated. We identified novel stop codons in positions of the *tjp1a* gene corresponding to the N-terminal part of the protein in all four new alleles. The phenotype is variable, and no qualitative differences between the alleles could be recognized. Individual fish of the *sbr*[tnh009b] allele with the C-terminal truncation may show a weaker phenotype not seen in the other alleles, therefore, we cannot exclude that it may have residual function. In subsequent crosses, we never observed a segregation of the *sbr* phenotype and the *tjp1a* mutant alleles. These results show that the loss of Tjp1a function causes the *sbr* phenotype.

## The *sbr* phenotype is not caused by a decrease in melanophore number

The larval pigment pattern is unaffected in *sbr* mutants (*Figure 2A*, 6.5 mm). Repeated photography of individual fish revealed that mutants can be distinguished from wild-type siblings at stage 7.5 mm SL (Standard Length [*Parichy et al., 2009*]) (about 4 weeks post fertilisation) shortly after the first metamorphic melanophores appear (*Figure 2A*). At this and following stages, melanophores in the mutants appear as small dots when compared to melanophores of wild type, giving the metamorphic fish a pale appearance (*Figures 2A*, 9.0–10.2 mm). Later (11 mm SL), the melanophores acquire

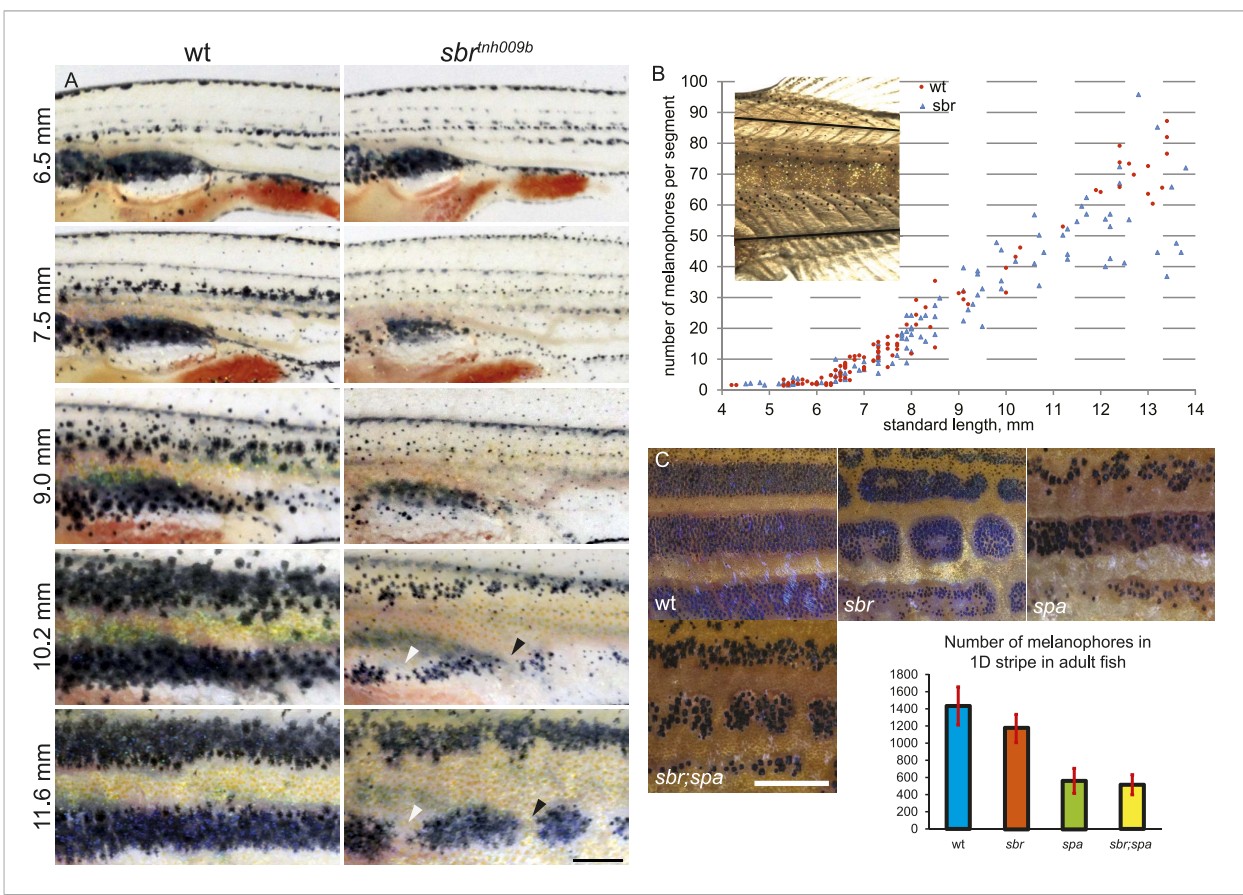

**Figure 2**. Abnormal behaviour of *sbr* mutant melanophores. (**A**) Pigment pattern during metamorphosis in the mid-trunk of individual wild type and *sbr* mutant fish. Arrowheads: forming interruptions. White arrowheads: disappearing melanophores (N = 6). Scale bar: 1 mm. (**B**) Average number of melanophores per segment in the first two dark stripes in wild type and mutant fish plotted against standard length. Red circles—individual wild type fish; blue squares—individual *sbr* fish. Inset shows the area where melanophores were counted. Distributions of melanophore numbers in mutants and wild type fish do not differ significantly until the 10 mm stage as shown by Kolmogorov–Smirnov statistics. At 10–14 mm stages the distributions are different with p-values < 0.05. (**C**) Close-ups of mid-trunk regions of adult wild type, *sbr*, *spa* and *spa;sbr* and melanophore numbers in a dark stripe dorsal to the first light stripe of adult fish. Red lines—standard deviation. Scale bar: 2 mm.

The following figure supplement is available for figure 2:

**Figure supplement 1**. Width of the first light stripe in *sbr* and wild type fish.

a shape similar to wild-type cells (*Figure 2A*, 11.6 mm). The melanophore numbers in mutant and wild-type fish do not differ significantly until 10 mm SL (*Figure 2B*), when the pale phenotype is already established. In older mutant fish, there is a slight decrease in the average number of melanophores, likely due to the interruptions of the stripe areas (*Figure 2C*, wt, *sbr*). To assess the impact of melanophore number on stripe integrity, we compared *sbr* to *sparse* (*spa*) mutants, which have decreased numbers of melanophores (*Johnson et al., 1995*). *spa* mutants have only about a third as many melanophores as wild-type fish (*Figure 2C*, plot); however, these cells form uninterrupted stripes (*Figure 2C*, *spa*). Double mutants *sbr;spa* display a combination of both phenotypes (*Figure 2C*, *sbr;spa*). This indicates that the pale appearance of the mutant metamorphic fish is caused by an abnormal size, shape, or pigment arrangement rather than a reduced number of melanophores.

## *sbr* iridophores fail to undergo shape change during early stripe formation

In early metamorphic mutant fish, but not in adults, the width of the first light stripe, composed of dense iridophores covered by compact yellow xanthophores, is increased compared to wild type (*Figure 2A*, 11.6 mm; *Figure 2—figure supplement 1*).

After 10 mm SL, dense S-iridophores and xanthophores can be observed in the dark stripe region in *sbr* mutants (*Figure 2A*, arrowheads) and melanophores disappear from these areas (*Figure 2A*, white arrowheads), ultimately leading to the interruptions. To investigate iridophore behaviour, we performed repeated imaging of wild-type and *sbr* individuals over a period of 2 weeks. To allow a more detailed visualisation of the cell shapes, we imaged fish carrying the *Tg(TDL358:GFP)* transgene (labelling iridophores and glia with cytosolic GFP [*Levesque et al., 2013*]) alone (*Figure 3*) or together with a second transgene, *Tg(sox10:mRFP)* (*Figure 4*), which labels neural crest derivatives with membrane-bound mRFP. In both, wild type and mutants, iridophores appeared in segmental clusters during early metamorphosis (about 7 mm SL), they increased in number and merged to form the first light stripe (*Figure 3A*, *Figure 4A*). In wild type, iridophores proceeded to define the edge of the light stripe, there they delaminated and formed loose iridophores, which spread dorsally and ventrally over the dark stripe regions (*Figure 3A*, 8.9 mm; *Figure 4*; *Singh et al., 2014*). Dense iridophores occasionally spread too far from the horizontal myoseptum (*Figure 3B*, wt), but later formed sharp light stripe borders. However, in the mutants the dense iridophores did not delaminate but continued to spread over the metamorphic melanophores as a coherent sheet (*Figure 3A*, *sbr* 8.9 mm; *Figure 4*, *sbr*, 8.3 mm). At later stages, eventually some of them switched to the loose form (arrowheads in *Figure 3B*; *Figure 4A*) and occasionally seemed to disappear from the dark stripe regions at a time point, which coincided with expansion of melanophores (10.5 mm SL, *Figure 3—figure supplement 1*). When this retreat did not happen, the iridophores persisted in interruptions of the dark stripes (*Figure 2A*, 11.6 mm). The failure to precisely form the boundary between light and dark stripes might be a cause for another anomaly observed in *sbr* mutants: L-iridophores, which are restricted to dark stripe areas in wild type, were observed in light stripes of adult *sbr* mutants (*Figure 3—figure supplement 2*).

Analysing fish carrying the transgene *Tg(kita:GalTA4:UAS:mCherry)*, which labels melanophores (*Anelli et al., 2009*), we observed that in *sbr* mutants individual melanophores moved away from invading dense iridophores, while maintaining a migratory stellate shape, or they disappeared after being trapped (*Figure 5*, *Figure 5—figure supplement 1*). This is in agreement with the observed reduction in the number of melanophores in *sbr* during later stages of development (*Figure 2B*).

## Tjp1a is required in iridophores for pattern formation

To investigate in which cell type *sbr* function is required, we analysed *sbr* in combination with mutants lacking one of the three pigment cell types. Both *shady* mutants, lacking iridophores (*shd*, *Figure 6C*, *Figure 6—figure supplement 1A*) and *shd;sbr* double mutants (*Figure 6D*, *Figure 6—figure supplement 1B*), display the *shd* phenotype with no detectable differences, suggesting that *sbr* function is only required in iridophores. In contrast, the phenotypes of double mutants with *nacre* (*nac*, no melanophores, *Figure 6E*) or *pfeffer* (*pfe*, no xanthophores, *Figure 6G*) differ from the single mutants. Both *pfe* and *nac* alone exhibit expanded areas of dense iridophores. In combination with *sbr*, both double mutants show a further expansion of these dense iridophore regions (*Figure 6F,H*), covering most of the body. This phenotypic enhancement suggests that the cell type affected in *sbr* is

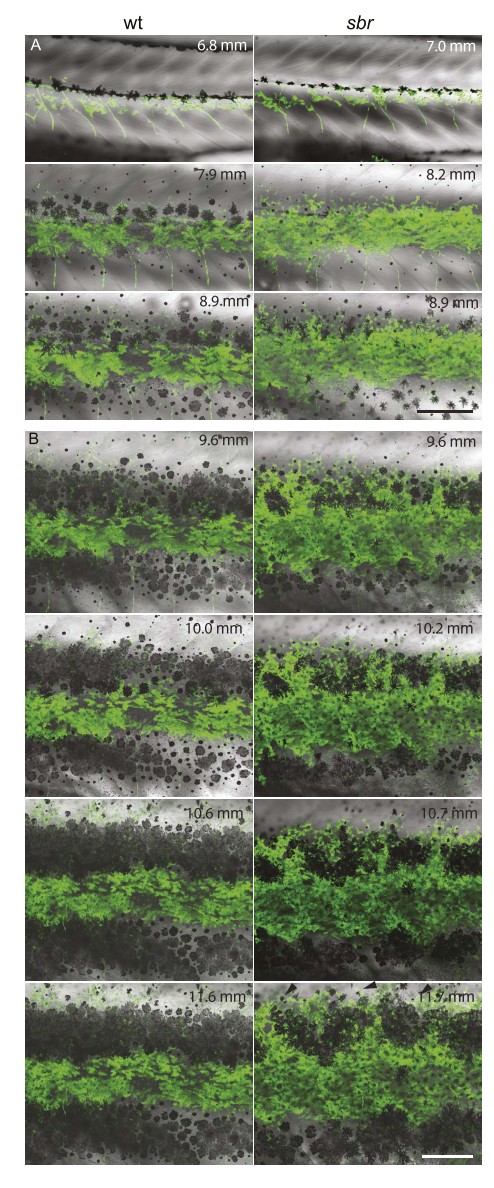

**Figure 3**. Behaviour of *sbr* mutant iridophores during metamorphosis. (**A**) Repeated imaging of *Tg(TDL358: GFP)* wild type and mutant metamorphic individual (N = 5 each, one shown). Scale bar: 300 μm. (**B**) Same individuals with another magnification. Empty patches in the light stripe of wild type fish are caused by variegation of the transgene expression. Arrowheads: loose iridophores. Scale bar: 300 μm.

The following figure supplements are available for figure 3:

**Figure supplement 1**. Invading *sbr* iridophores occasionally retreat.

**Figure supplement 2**. L-iridophores in wt and *sbr*.

still present in *nac* and *pfe* mutants, again pointing to iridophores. To confirm these findings, we created chimeric animals by blastomere transplantations. Experiments with *sbr* donors and *nac* or *pfe* recipients revealed that *sbr* melanophores and xanthophores can participate in normal pattern formation (*Figure 6I,J*). When we used *shd*;*sbr* double mutants as recipients (*Figure 6D*) and *nac*;*pfe* (*Figure 6K*) as donors, which can provide only iridophores, we observed regional restoration of the striped pattern in the chimeric fish (*Figure 6L*). This indicates that *sbr* is required cell autonomously in iridophores and confirms that mutant *sbr* melanophores and xanthophores can contribute to the normal pattern when confronted with wild-type iridophores.

The fins of sbr mutants are striped, although we detect branching and supernumerary stripes to various extents in the caudal fins of some *sbr* mutant fish but not in their anal fins suggesting that there is no systematic defect in fin patterning. This is in agreement with the finding that iridophores are not required for striping the fins (*Hirata et al., 2005*; *Frohnhöfer et al., 2013*; *Krauss et al., 2013*).

## Tjp1a is expressed in dense iridophores but not in loose iridophores nor other pigment cells

We raised two polyclonal antibodies in rabbits specific to zebrafish Tjp1a (*Figure 1D*). α-Tjp1aN was designed to recognize both, truncated *sbr^{tnh009b}* and wild-type Tjp1a protein, whereas α-Tjp1aC would only bind to the wild-type protein. Both antibodies allow the detection of Tjp1a in epithelial cells of larval and adult zebrafish skin (*Figure 7—figure supplements 1, 2*). This staining is absent in mutants with stop codons in the N-terminal part of *tjp1a* but present in *sbr^{tnh009b}* mutants stained with α-Tjp1aN (*Figure 7—figure supplement 1*). We also detected expression of Tjp1a in blood vessels during larval and adult stages (*Figure 7—figure supplement 2*), corroborating earlier reports on the expression of Tjps in zebrafish and mice (*Anderson and Itallie, 1995*; *Blum et al., 2008*). Immunostaining of skin in metamorphic fish carrying the *Tg(TDL358:GFP)* transgene (*Figure 7A*) shows that Tjp1a is expressed in dense iridophores of the light stripe. Intriguingly, delaminated loose iridophores still express GFP, but no Tjp1a is detectable (*Figure 7B*). This indicates that Tjp1a is down-regulated during delamination of loose iridophores from the dense

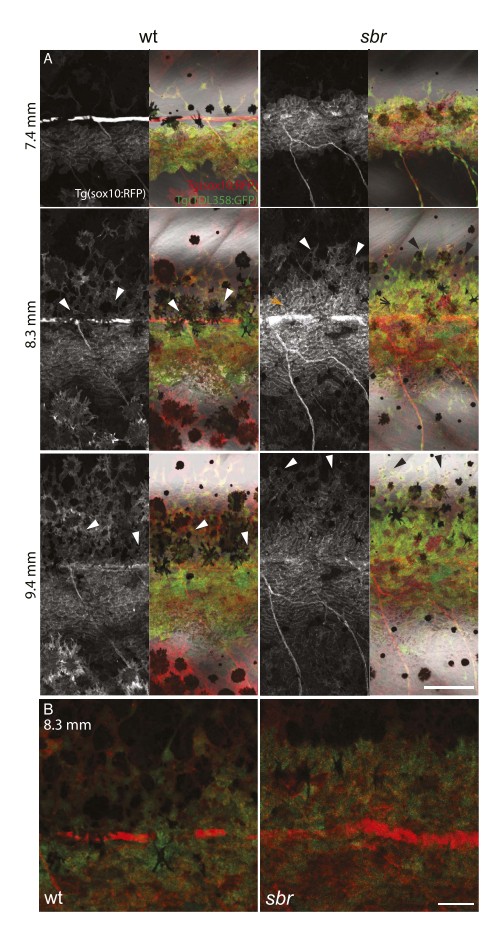

**Figure 4**. Behaviour of *sbr* mutant iridophores during establishment of the first dark stripes. (**A**) *Tg(TDL358: GFP); Tg(sox10:mRFP)* wild type and *sbr* metamorphic fish (N = 4 each, one shown). Arrowheads point to delaminating loose iridophores. Arrow shows dense iridophores failing to delaminate. Scale bar: 150 μm. (**B**) Close-ups of *Tg(TDL358:GFP); Tg(sox10:mRFP)* wild type and *sbr* metamorphic fish 8.3 SL. Note difference in iridophore shapes in wild-type. Scale bar: 50 μm.

sheet in the light stripe. In adult skin preparations, the signal can be observed in dense iridophores of the light stripes but not in xanthophores, melanophores, L-, or loose iridophores (*Figure 7C*). Together with our observation that the loss of *tjp1a* function in *sbr* mutants compromises the transition of iridophores from dense to loose state, these results suggest that Tjp1a is a component of the molecular switch that regulates iridophore shape changes during their dispersal.

Additionally, we analysed chimeras obtained by transplanting blastomeres from *sbr^twl4^* embryos, where transplanted cells were labelled with expression of the ubiquitous *Tg(H2A:GFP)* transgene, into blastula stage wild-type embryos. Double stainings with α-Tjp1aN and α-GFP antibodies show that the donor-derived *sbr* dense iridophores integrate with the wild-type recipient iridophores but do not express Tjp1a (*Figure 7D*). This suggests that the *sbr* phenotype is not caused by over-proliferation of iridophores, since they do not produce large clusters. We stained skin of *nac;pfe/shd;sbr* chimeras with α-Tjp1aC and detected Tjp1a in donor-derived iridophores but not in the epithelium, suggesting that loss of Tjp1a function in the epithelium does not affect pattern formation (*Figure 7—figure supplement 3*).

## *sbr* enhances connexin mutant phenotypes

To investigate the genetic interactions between *tjp1a* and potential partners, *cx39.4* and *cx41.8*, we evaluated the phenotypes of double mutants with *luc^t32241^* and *leo^t1^* (*Figure 8*). *luc* mutant fish display meandering and broken stripes, whereas in *leo^t1^* the stripes are broken into spots. In the double mutants with *sbr*, we observe considerably stronger patterning defects than in the single mutants. In the case of *sbr;luc*, an almost complete loss of melanophore clustering is observed; the upper part of the body is covered with a layer of dense iridophores. In the case of *sbr;leo*, the melanophore spots are even smaller and the dense iridophore-free areas around them are narrower. These results suggest that connexins and *tjp1a* do not act in a linear pathway affecting pigmentation. To investigate whether zebrafish Tjp1a can interact directly with connexins, we performed yeast two-hybrid assays (*Figure 8—figure supplement 1*). We observed interactions between Cx41.8 and all three PDZ domains of Tjp1a and between Cx39.4 and PDZ-2 and 3 in this assay.

## Discussion

We show that Tjp1a-deficient fish develop multiple interruptions of the dark stripes in the trunk by light stripe structures composed of dense iridophores covered by compact xanthophores. In *sbr* mutants, dense iridophores of the light stripe spread during metamorphosis as a coherent sheet invading dark stripes rather than loosening up and dispersing. Melanophore expansion into the stationary rounded form is temporarily suppressed. However, clustered melanophores later expand and seem to repulse iridophores in a process, which is similar to the one involved in smoothening of stripe boundaries

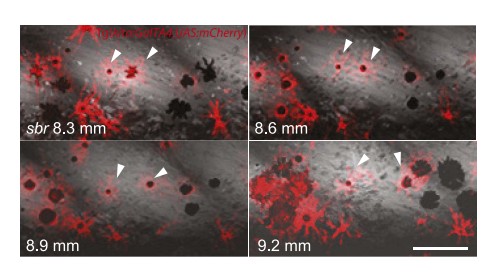

**Figure 5**. Two closely positioned melanophores in *sbr* (arrowheads), are migrating away from the iridophores in posterior and anterior directions. Scale bar: 100 μm.

The following figure supplement is available for figure 5:

**Figure supplement 1**. Melanophores trapped in the mass of iridophores are disappearing in *sbr*.

(*Frohnhöfer et al., 2013*; *Singh et al., 2014*). The location of the interruptions seems to be random. Genetic mosaics, double mutant analysis, as well as immunostaining indicate that Tjp1a is expressed and required in dense iridophores but not in melanophores or xanthophores. We show that, surprisingly, Tjp1a-deficient iridophores do display a dense shape and organisation, however, intriguingly, they fail to undergo the transition to the loose shape. This suggests that the cell shape of loose iridophores is not determined by the absence of Tjp1a per se. In contrast, a reduction in the levels of Tjp1a may be read by iridophores as a trigger for the transition or the cell shape change might result in a down-regulation of Tjp1a. In the absence of Tjp1a, other Tjps might take over the role in cell compaction but may not be able to properly respond to cues guiding the transition to the loose shape.

## Gene duplication and redundancy of Tjps functions in zebrafish

One surprising finding of this study is that Tjp1a-deficient zebrafish are viable unlike embryonic lethal Tjp1$^{-/-}$ mice (*Katsuno et al., 2008*). There are three *tjp* genes (1–3) in mammals and five in zebrafish (1a–b, 2a–b, 3), due to the whole genome duplication in teleosts (*Amores et al., 1998*). The lack of Tjp1a function in epithelial cells in *sbr* mutants might be compensated for by other Tjps, for example, Tjp1b, which does not exist in mammals. This is supported by the observation that morpholino-mediated knockdown of *tjp1b* in *sbr* mutants, but not wild-type embryos, results in impaired blood flow and death at 5 dpf (*Videos 1–3*). This suggests that Tjp1b and Tjp1a have redundant functions at least in the vasculature epithelial cells. This notion is supported by experiments with mammalian cell cultures showing that absence of ZO-1 leads to increased recruitment of ZO-2 to cell membranes, which is suggested to compensate for the absence of ZO-1 (*Umeda et al., 2004*).

## Tjp1a-induced cell shape transition during colour pattern formation

Our data show that in *sbr* dense iridophores fail to switch to the loose form in dark stripe regions. In wild type, dense iridophores normally stay restricted to developing light stripes, but occasionally spread into the prospective dark stripe areas. This irregularity is usually corrected and sharp stripe boundaries are formed. However, in *sbr*, the invasion of dense iridophores occurs along the whole length of stripes. Not all dense iridophores persist in dark stripe regions in *sbr* mutants. In summary, we hypothesize that the loss of Tjp1a impairs the ability of iridophores to recognise the (as yet unknown) cues defining the dark stripe areas or their ability to react to them efficiently. So far, only a rather small number of molecules have been identified, which are involved in the various interactions between chromatophores. Tjp1a is the first for which a molecular distribution and cell type specific expression has been shown.

## Tjp1a might interact with connexins/gap junctions

Several zebrafish mutants including *leopard* (*Haffter et al., 1996*; *Watanabe et al., 2006*), *luchs* (*Irion et al., 2014*), and *seurat* (*Eom et al., 2012*) exhibit a spotted pattern formed by ingressions of iridophores into the dark stripe area. *luc* and *leo* encode Connexin41.8 (Cx41.8) and Connexin39.4 (Cx39.4), respectively, which are, in contrast to *sbr*, required in melanophores and xanthophores (*Maderspacher and Nüsslein-Volhard, 2003*; *Irion et al., 2014*). Irion et al. suggest that Cx39.4 and Cx41.8 form heteromeric gap junctions, promoting interactions of melanophores and xanthophores that result in the appropriate patterning of iridophores. In the absence of xanthophores or melanophores, dense iridophore regions are expanded (*Frohnhöfer et al., 2013*), suggesting that Tjp1a in iridophores may be involved in cell communication with xanthophores and/or melanophores. However, the downstream cytoplasmic partners of the transmembrane proteins shown to be involved in patterning in melanophores and xanthophores are unknown as well as transmembrane molecules in

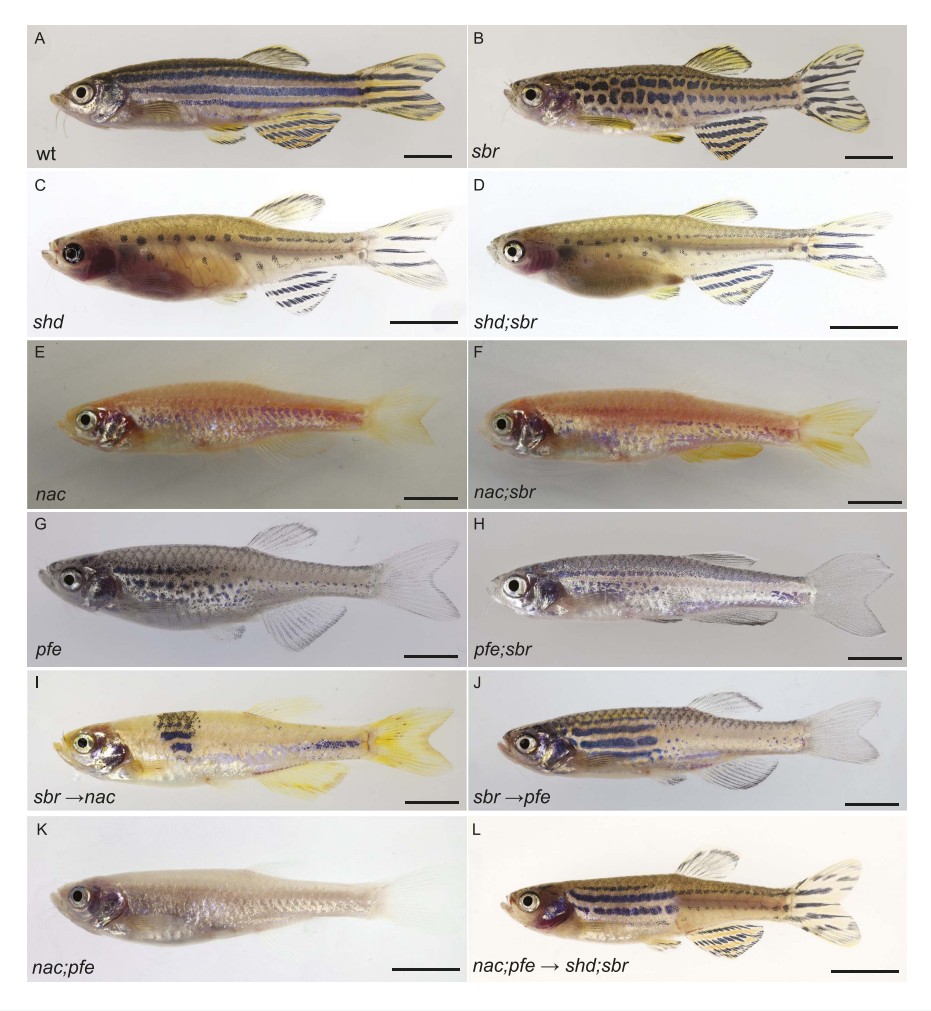

**Figure 6**. *tjp1a* is required in iridophores, but not melanophores or xanthophores. (**A**) Wild type fish. (**B**) *sbr* fish. (**C**) *shady* (*shd*) mutant, which lacks iridophores. (**D**) *shd*;*sbr* mutant is indistinguishable from *shd*. (**E**) *nacre* (*nac*) mutant, which lacks melanophores. (**F**) *nac*;*sbr* double mutant exhibiting expanded dense iridophore areas in comparison to *nac* alone. (**G**) pfeffer(*pfe*) mutant, which has no xanthophores. (**H**) *pfe*;*sbr* double mutant exhibiting expanded dense iridophore areas in comparison to *pfe* alone. (**I**) Chimeras, obtained from transplantation of *sbr* blastomeres into *nac* recipient blastulas, show clonal rescue. (**J**) Chimeras obtained from transplantation of *sbr* blastomeres into *pfe* recipient blastulas, show clonal rescue. (**K**) *nac*;*pfe* fish have only one type of pigment cells—iridophores. (**L**) Chimeras obtained from transplantation of *nac*;*pfe* blastomeres into *shd*;*sbr* recipient blastulas, show clonal rescue. Scale bars: 5 mm.

The following figure supplement is available for figure 6:

**Figure supplement 1**. Phenotypes of *shd* and *shd*;*sbr* mutants.

iridophores that are responsible for the interactions. Another mutant displaying a spotted pattern is *seurat*, encoding the transmembrane protein immunoglobulin superfamily member 11 (Igsf11) (*Eom et al., 2012*). Interestingly, Cx41.8 and Igsf11 are possible interacting partners of Tjps since they have putative PDZ-binding motifs on their extreme C-termini (*Hung and Sheng, 2002*; *Suzu et al., 2002*). The multiple protein–protein interacting domains in Tjps allow for many interacting partners and facilitate formation of large complexes in proximity of cell membranes that are associated with tight, adherens, and gap junctions. These provide a link between transmembrane proteins and the cytoskeleton and were shown to participate in regulation of many cellular processes such as junction assembly, cell proliferation, and differentiation (*Balda and Matter, 2000*; *Bauer et al., 2010*;

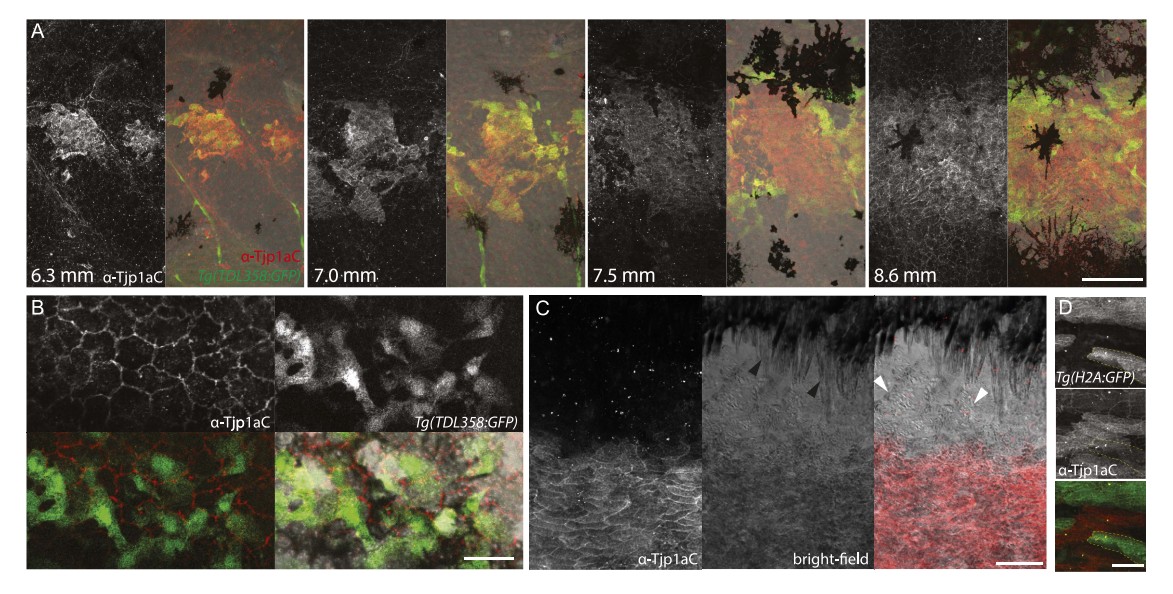

**Figure 7**. Tjp1a is expressed in dense iridophores. (**A**) Double antibody staining of metamorphic *Tg(TDL358:GFP)* fish with α-Tjp1aC and α-GFP antibodies. Note: not all iridophores are expressing GFP due to transgenic line variegation. Scale bar: 100 µm. (**B**) Loose iridophores migrating over the dark stripe in 8.3 mm metamorphic *Tg(TDL358:GFP)* fish express GFP, but not Tjp1a, although the epithelial staining is still visible. Scale bar: 30 µm. (**C**) α-Tjp1aC staining in skin of adult wild type fish. The protein is detected in the sheet of dense S-iridophores of the light stripe, but not in L-iridophores (black arrowheads), loose iridophores (white arrowheads), melanophores or xanthophores. Scale bar: 100 µm. (**D**) Double antibody staining with α-Tjp1aC and α-GFP of skin of adult chimera, obtained by transplanting *sbr;Tg(H2A:GFP)* blastomeres into wild type blastula. Either GFP or Tjp1a was detected in cells, never both. Some *sbr* cells express no GFP due to variegation of the transgene expression. Scale bar: 30 µm.

The following figure supplements are available for figure 7:

**Figure supplement 1**. Tjp1a stainings in wild type and *sbr*.

**Figure supplement 2**. Characterization of the Tjp1a expression domain.

**Figure supplement 3**. Correlation between clonal rescue of *sbr* phenotype and Tjp1a expression.

*Xu et al., 2012*; *González-Mariscal et al., 2014*). Our results show that *sbr* enhances the phenotypes of both *luc* and *leo* mutants. This suggests that Tjp1a and connexins do not act in a linear pathway to regulate pattern formation, but most likely work through different mechanisms. One possible explanation is that Tjp1a is required for spatially and temporally controlled reaction of iridophores in response to melanophores (directly or through xanthophores). Absence or truncation of Tjp1a results in a delayed switch to the loose form, which in turn forces melanophores to reorganize according to the presence of dense iridophores in normally iridophore-free regions. In *luchs* and *leopard*, the melanophore and xanthophore autonomous mutations also affect patterning of iridophores, likely due to the failure to properly guide iridophores (*Irion et al., 2014*). The combined effect of failure of melanophores and xanthophores to provide cues to iridophores, and the delayed reaction of iridophores might be responsible for the enhanced phenotypes in the double mutants.

Interestingly, Tjp1a and Cx39.4 and Cx41.8 can interact in a yeast 2-hybrid assay. *As in the fish they are required in different pigment cell types,* this may point to the existence of other, as yet unknown, connexins, similar to the Cx41.8 and Cx39.4, that are expressed in iridophores and interact with Tjp1a. ZO-1 was shown to regulate gap junction assembly, localization, and regulate plaque size in mammalian cell cultures (*Hunter et al., 2005*; *Laing et al., 2005*; *Rhett et al., 2011*). Defective Tjp1a in *sbr* might affect proper interaction of iridophore connexins with their counterparts in melanophores and xanthophores, compromising cell–cell communication and recognition.

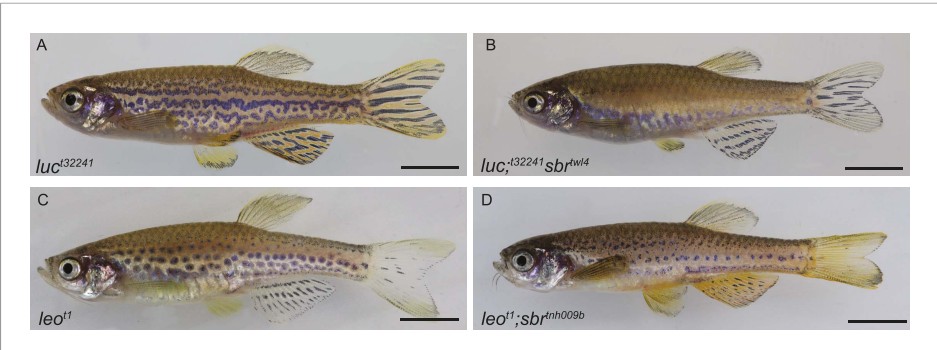

**Figure 8**. Genetic interactions between *luc*, *leo* and *sbr*. (**A**) *luchs^{t32241}* (*luc*) mutant affects Cx39.4 and results in meandering and broken stripes. (**B**) *luchs^{t32241};sbr^{twl4}* mutant exhibits complete loss of stripes and expansion of dense iridophore area. (**C**) leopard^{t1} (*leo*, cx41.8) stripes are broken into spots. (**D**) *leo^{t1};sbr^{tnh009b}* double mutant displays decrease in the size of the spots. Scale bars: 5 mm.

The following figure supplement is available for figure 8:

**Figure supplement 1**. Interaction of PDZ domains of Tjp1a with connexins.

## Tjp1a as a regulator of cell shape

Iridophore-specific connexins or other molecules, responsible for communication between pigment cells, might also transmit signals via Tjp1a, controlling iridophore migration or shape change in a spatiotemporally appropriate manner. Immunostainings show that Tjp1a is expressed in dense iridophores, but not in loose iridophores. Intriguingly, the absence of Tjp1a does not obviously affect the morphology of dense iridophores, which display normal shape and organisation. In vitro studies of the past decade have demonstrated a function of ZO-1 in organisation of confluent cell layers. Counterintuitively, ZO-1 −/− Eph4 cells polarize and form tight junctions morphologically indistinguishable from those of ZO-1 +/+ cells, but the formation is delayed. These cells do not exhibit abnormal growth or motility in scratch assays (*Umeda et al., 2004*). However, knockdown of endogenous ZO-1 in COS-7 cells hampers delamination and migration of cells to fill the wound area in scratch assays (*Huo et al., 2011*). These data suggest that epithelial cells of different origin may react differently to the absence of ZO-1. Our Tjp1aN antibody shows that the non-functional truncated protein is at least partially retained and normally localized in *sbr^{tnh009n}* mutants, suggesting that the missing domains (ZU-5 and possibly parts of afadin- and actin-binding regions [*Bauer et al., 2010*]) are crucial for the function of Tjp1a in iridophores. It was shown that absence of the ZU5 domain of ZO-1causes defective delamination and migration of COS-7 cells (*Huo et al., 2011*). Furthermore, mis-expression of truncated ZO-1 in the presence of the wild-type protein in CE culture leads to the expression of mesenchymal markers and to an epithelial–mesenchymal transition (EMT) (*Ryeom et al., 2000*). Taken together with our findings, these data suggest a role of ZO-1 in regulating and fine-tuning of cell shape and state.

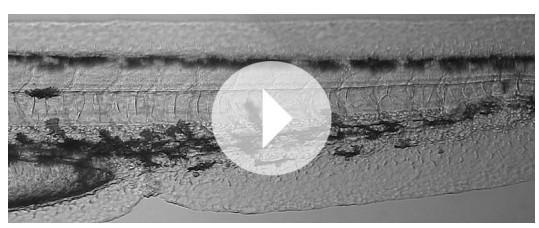

**Video 1.** 50 hpf wild type embryo. Note normal blood flow.

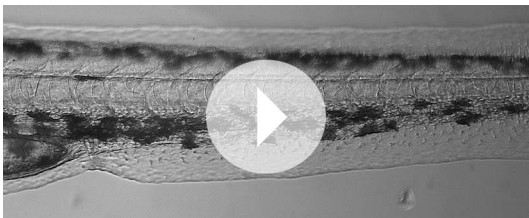

**Video 2.** 50 hpf wild type embryo injected with morpholino against Tjp1b. Note normal blood flow (N = 53/53). The result shows non-toxicity of morpholino. No defects are observed in the injected fish (observed until adulthood).

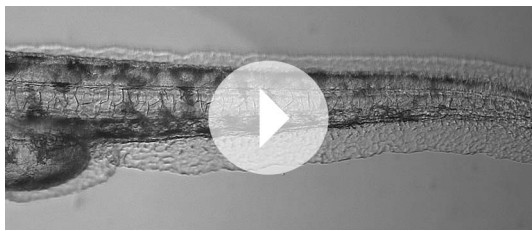

**Video 3.** 50 hpf *sbr* embryo injected with morpholino against Tjp1b. No blood flow is observed, possibly due to disrupted angiogenesis (N = 30/38). None of the 30 individuals without blood flow survived past 5 dpf.

We show that transitions in cell shape and organisation are crucial for the arrangement of pigment cells in stripes and identify Tjp1a as a regulator of this process. It appears that the presence of Tjp1a allows iridophores to change into the loose form at the appropriate positions. This suggests that Tjp1a is required for interaction of iridophores with other pigment cells (for example through controlling assembly of gap junctions) and/or appropriate reaction of iridophores to perceived cues (through control of delamination and cell shape). The spatial and temporal regulation of iridophore shape transitions by Tjp1a might underlie the generation of a variety of patterns observed in teleosts. Moreover, the viability of *sbr* mutants presents exciting opportunities for studying the behaviour of Tjp1 deficient cells in vivo.

## Materials and methods

### Zebrafish maintenance
Fish were bred and maintained as described (*Nüsslein-Volhard and Dahm, 2002*). Fish of the following genotypes were used: Tü, WIK, TE wild-type strains (Tübingen zebrafish stock centre), *luchs*[t32241](*Irion et al., 2014*), *leo*[t1] (*Watanabe et al., 2006*), *nacre*[w2] (*Lister et al., 1999*), *pfeffer*[tm236b] (*Odenthal et al., 1996*), *shady*[j9s1] (*Lopes et al., 2008*), *sparse*[b134] (*Kelsh et al., 1996*), Tg(TDL358: GFP) (*Levesque et al., 2013*), Tg(kdrl:GFP) (*Jin et al., 2005*), Tg(kita:GalTA4:UAS:mCherry) (*Anelli et al., 2009*), Tg(sox10:mRFP) (M Levesque; CN-V laboratory), Tg(H2A:GFP) (A Mongera; CN-V laboratory). Fish were staged according to the normal table of zebrafish development (*Parichy et al., 2009*).

### Mutagenesis
The original allele *sbr*[tnh009b] was identified in a screen for mutants induced with N-ethyl-N-nitrosourea (N5809, Sigma-Aldrich, St. Louis, Missouri) in Tü wild-type background. Mutagenesis was carried out as described previously (*Rohner et al., 2011*). Subsequently, fish were crossed to TE and later maintained in homozygosity by regular outcrossing. Four new alleles were isolated by crossing mutagenized Tü males to *sbr*[tnh009b] females and screening the adult progeny for the *sbr* phenotype.

### Mapping and alleles testing
*sbr*[tnh009b]/WIK fish were incrossed and used for meiotic mapping as described previously (*Nüsslein-Volhard and Dahm, 2002*). The mutation was mapped to the region between microsatellite markers z4706 (36.7 cM) and z52932 (41.4 cM) on chromosome 7. Further, the interval was narrowed to the region 29.6–32.5 Mb of chromosome 7, between contigs CR356242 and BX3235912 (Ensembl Zebrafish release 72). The following primers were used:

CR356242_F GTAGTATATGGATATGGATG
CR356242_R CCACCGCTGCATACCCTGC
BX3235912_F CTTGCACAGGGAATGTGT
BX3235912_R CTGCAGTGTTCTCACGCT

To check for presence of lesions in *tjp1a,* RNA was extracted from blastema of adult wild-type and *sbr* fish using TRIzol reagent (15596, Thermo Fisher Scientific, Waltham, Massachusetts). cDNA was obtained using Omniscript RT kit (205111, Qiagen, Netherlands). Four overlapping regions of the coding region of *tjp1a* (ENSDART00000148347) were amplified using Taq polymerase S (M3001.0250, Genaxxon, Germany) and the following primers:

tjp1a_1F 5′-GACTGCGGGATTTCAGTTGT-3′
tjp1a_1R 5′-CACTATTCGCCGGTACACATC-3′
tjp1a_2F 5′-GCAGAAGAAGAAAGATGTGTAC-3′

tjp1a_2R 5′-ATGTGAACCGTCCGCCTTG-3′
tjp1a_3F 5′-CAACCATCATCTCTTCACAGCCACT-3′
tjp1a_3R 5′-GATTTTCTCCACTGACTCTGCTCTGG-3′
tjp1a_4F 5′-CTGGATCAAGAGAAGACCTTTAGAACTC-3′
tjp1a_4R 5′-TCCCTGCAGTCTCAGAGGTT-3′.

PCR products were cloned into pGEM-T Easy (A360, Promega, Fitchburg, Wisconsin) and sequenced using Big Dye Terminator v3.1 kit (4337455, Thermo Fisher Scientific).

## Generation of polyclonal antisera

Two parts of the *tjp1a* cDNA corresponding to 992–1143 a.a. (α-Tjp1aN) and 1293–1397a.a. (α-Tjp1aC) of Tjp1a (ENSDART00000148347) were cloned both into pET28-nusA (Novagen) and pOPT-GST-Kan (gift from U Irion and O Perisic) plasmids to produce 6xHis-nusA and GST-tagged fusions. The following primers were used to amplify these regions:

tjp1aN_F 5′-CATATGTACAAGAAGGATATCTACCGACCC-3′
tjp1aN_R 5′-GGATCCTTAGGAAGGCCTTTGGG-3′
tjp1aC_F 5′-CATATGAAACCCTCCACACAGCTGACAC-3′
tjp1aC_R 5′-GGATCCTTAGCTGGACGTGGCAG-3′.

Obtained constructs were used to transform BL21-CodonPlus DE3-RIPL (230280, Agilent Technologies, Santa Clara, California) cells. The cells were grown in 1 ml of 2xTY medium containing 20 mM glucose and 15 µg/ml kanamycin for 3 hr on 37°C, 220 rpm. This culture was used to inoculate 50 ml of the same medium and was grown overnight on 20°C, 220 rpm. His-tagged polypeptides were purified using HiTrap IMAC FF 1 ml (17-0921, GE Healthcare, UK) charged with $Ni^{2+}$ and 250 mM imidazole in the elution buffer. GST-tagged polypeptides were purified using GSTrap FF 1 ml (17-5130, GE Healthcare). In all cases, the samples of eluted proteins were loaded on NuPage Novex 4–12% Bis-Tris gel (NP0322BOX, Thermo Fisher Scientific) and stained with Coomassie Brilliant Blue G-250 to assess the purity. The polypeptides were dialyzed in PBS using Slide-A-Lyzer Dialysis Cassettes 10K MWCO (66383, Thermo Fisher Scientific). The protein concentrations were assessed using Bradford method. His-tagged polypeptides were used to immunize rabbits with Freund's complete adjuvant (F5881, Sigma-Aldrich) as immunopotentiator. GST-tagged polypeptides were bound to HiTrap NHS-activated HP columns (17-0716, GE Healthcare) and used to purify the corresponding antibodies from rabbit serum, using PBS as binding buffer and 100 mM glycine pH 2.3 as elution buffer. The purified antibodies were neutralized with Tris-HCl pH 9.5 and mixed 1:1 with glycerol.

## Immunohistochemistry

Antibody staining was performed as described previously (*Singh et al., 2014*) omitting methanol hydration/rehydration and HCl steps. Antibodies used were mouse α-E-cadherin (610181, BD Biosciences, Franklin Lakes, New Jersey), mouse α-GFP (11814460001, Roche, Germany), goat α-rabbit coupled with Cy3 (111-165-003, Dianova, Germany), goat α-mouse AlexaFluor 488 (A21131, Molecular Probes, Eugene, Oregon). All antibodies were used in 1:400 dilution, except α-Tjp1aN and α-Tjp1aC, which were used in 1:100 dilution.

## Transplantations

Chimeras were produced as described (*Nüsslein-Volhard and Dahm, 2002*) using mid-blastula stage (1000 cell stage) embryos, transplanting 30–60 cells.

## Image acquisition

We used Zeiss LSM 780 NLO confocal microscope and Canon 5D Mk II camera to obtain images. Fiji (*Schindelin et al., 2012*), Adobe Photoshop, and Adobe Illustrator CS6 were used for image processing and analysis. Maximum intensity projection was made for fluorescent channels of confocal scans. For bright-field images, we used 'stack focuser' plugin or a single slice on an appropriate depth. For adult fish photos, multiple RAW camera images were taken in different focal planes and auto-align and auto-blend functions of Photoshop were used. Repeated imaging of metamorphic fish and anaesthesia were performed as described previously (*Singh et al., 2014*).

## Melanophore counts

Melanophores in metamorphic fish were counted in five segments in the middle 70% of myotome starting with the one above the first ray of the anal fin and proceeding posteriorly.

A Kolmogorov–Smirnov test was conducted in SciPy (*Jones et al., 2001*) to compare the distributions of melanophore counts in mutant vs wild-type fish. An initial comparison was conducted on fish of 4–6 mm SL. Sample sizes were then increased to include the melanophore counts of fish of 6–7 mm SL, and each subsequent data set was formed in a similar fashion by 1 mm increment. The null hypothesis of the samples being drawn from the same distribution was rejected with a p-value of 0.011 when a data set composed of 4–10 mm SL fish was used.

### Light stripe width quantification

For measuring the first light stripe width, the light stripe was defined as an area taken by dense iridophores. The width of the stripe was measured along five lines, perpendicular to the lateral line and drawn from the bases of each second fin ray in the anal fin starting with the first. The body height was measured along the first line.

### Morpholino injections

The knockdown was performed as described before (*Nüsslein-Volhard and Dahm, 2002*) using 3 ng of *tjp1b*-MO (CGAGTATGTGATCAGTCTTACTGCA), obtained from Gene Tools, LLC, Philomath, Oregon.

### Yeast two-hybrid assay

The PDZ domains of ZO-1 were amplified by RT-PCR from wild-type zebrafish RNA with the following primer pairs:

T878: 5′-CATATGGTGACTCTTCACAGGGCACC-3′
T879: 5′-GGATCCTTCCGCTTCCTGCGGATAG-3′
T880: 5′-CATATGGTCACACTCGTCAAGTCCCGC-3′
T881: 5′-GGATCCTTCATCTCTCTGCACCACCAT-3′
T882: 5′-CATATGAAGTTTAAGAAAGGGGAAAGTG-3′
T883: 5′-GGATCCTTTCTTCTTCTGCGCAAGGATGG-3′

and cloned in the vector pGBKT7 (Takara, Japan) via NdeI and BamHI.

Similarly, the C-termini of Cx39.4, Cx41.8, and Cx43 were amplified by RT-PCR with the following primer pairs:

T886: 5′-CATATGCTTCAGTTGGTGATAAC-3′
T887: 5′-GGATCCTCAAACATAATGTCTCGGTTTG-3′
T884: 5′-CATATGGCATGGAAGCAGTTGAGG-3′
T885: 5′-GGATCCTATACCGCAAGGTCGTCCGG-3′
T888: 5′-CATATGCTCTTCAAACGAATCAAGGACC-3′
T889: 5′-GGATCCTAGACGTCCAGGTCATCAGG-3′

and cloned into the vector pGADT7 (Clontech) via NdeI and BamHI.

The plasmids were transformed into the yeast strain Y2HGold (Clontech) by standard procedures, and we screened for positive interactions using X-α-Gal and His as markers.

## Acknowledgements

We thank A Mongera and M Levesque for providing transgenic lines; A Singh for discussions, expertise, and valuable comments on the manuscript; M Sonawane and D Gilmour for discussions and comments on the manuscript; O Perisic for providing plasmids; R Neher for help in melanophore count statistics. We thank H-M Maischein for help with blastomere transplantations, C Liebig for support in light microscopy, B Walderich and the fish facility, W Antonin and the rabbit facility for great support. This work was financially supported by the Max-Planck Society for the Advancement of Science and by ZF-HEALTH grant (Project number 242048).

## Additional information

### Funding

| Funder | Grant reference | Author |
| --- | --- | --- |
| Max-Planck-Gesellschaft (Max Planck Society) | | Christiane Nüsslein-Volhard |

| Funder | Grant reference | Author |
|--------|-----------------|--------|
| European Commission (EC) | ZF-HEALTH grant (Project number 242048) | Christiane Nüsslein-Volhard |

The funders had no role in study design, data collection and interpretation, or the decision to submit the work for publication.

### Author contributions

AF, Conception and design, Acquisition of data, Analysis and interpretation of data, Drafting or revising the article; JK, Identification of twl4, twl1, twc2 alleles, Conception and design, Analysis and interpretation of data; HGF, Isolated tnh009b allele; UI, Acquisition of data, Analysis and interpretation of data, Drafting or revising the article; CN-V, Conception and design, Analysis and interpretation of data, Drafting or revising the article

### Ethics

Animal experimentation: All animal experiments were performed in accordance with the rules of the State of Baden-Württemberg, Germany. The protocol for ENU mutagenesis was approved by the Regierungspräsidium Tübingen (Aktenzeichen: 35/9185.81-5/Tierversuch-Nr. E 1/09).

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
