## [Decision Letter]

Thank you for sending your work entitled “Tight junction protein 1a regulates pigment cell organisation during Zebrafish colour patterning” for consideration at *eLife*. Your article has been favorably evaluated by Diethard Tautz (Senior editor), Marianne Bronner (Reviewing editor), and three reviewers.

The Reviewing editor and the reviewers discussed their comments before we reached this decision, and the Reviewing editor has assembled the following comments to help you prepare a revised submission.

The authors study the functions of the tight junction protein, Tjp1a, in pigment cell development in zebrafish. Multiple mutant alleles of Tjp1a clearly demonstrate an essential function in metamorphic stripe formation. Loss of Tjp1a leads to ectopic spreading of dense iridophores and interruptions in melanophore stripes. Both expression of Tjp1a in iridophores and the ability of wild-type iridophores to restore stripe formation in Tjp1a mutants indicate a function in iridophores for interactions with other pigment cell types. Identification of a novel role for a tight junction protein in pigment cell development is interesting and suggests similar roles in cell-cell communication in other contexts. However, the paper requires further discussion and experiments to make the results more mechanistic and less descriptive. In particular, it is critical to add experiments to show an interaction between Tjp1a and the connexins.

Major comments:

1) Several previously published mutants (*luc*, *leo*) with similar pigment stripe defects disrupt connexins associated with gap junctions in pigment cells (though not specifically in iridophores), and in the Discussion the authors speculate that these may interact with Tjp1a. However, there are no experiments in the paper to address this hypothesis, such as tests of genetic interactions in *sbr*;*luc* and *sbr*;*leo* double mutants, evidence for physical interactions between proteins, or assays for functional gap junctions. Any evidence for (or against) the idea that loss of Tjp1a function disrupts gap junctions would greatly increase the paper's impact. The authors briefly mention that a reduction in Tjp1a may trigger iridophores to undergo the transition from dense to loose and more migratory behaviors. Would gap junctional communication likely influence this transition and if not what alternative types of cell-cell interactions might be involved?

2) Much of the data is shown at low magnification and lacks cellular resolution: In Figure 2 melanophores appear smaller in Tjp1a mutants, which could account for altered behaviors of iridophores. The authors argue instead that this is simply a difference in melanin distribution within melanophores of the same size, but this cannot be seen in the images provided. Mutant melanophores also appear smaller in Figure 4. Individual iridophores/xanthophores are also difficult to see in these figures.

Interpretations of iridophore behaviors depend on the ability to follow individual cells, such as TDL358:GFP+ cells in Figure 4. The authors claim to have performed time-lapse imaging in the Results but do not appear to have tracked individual cells based on the still images provided and do not explain their time-lapsing procedures in the Methods. If not, some apparent cell movements may instead reflect changes in transgene expression or iridophore differentiation.

The supposed lack of immunostaining for Tjp1a in loose iridophores in Figure 7 is unclear and difficult to distinguish from protein localized in surrounding epithelial cells.

3) The information about the ultimate mechanism is a bit speculative. The phenotype should be corroborated using CRISPR technology.

4) Previous work from the same group published in *eLife* (18) provided a genetic basis for a similar spotted phenotype. In this case the underlying connexin genes were required in the melanophores and xanthophores, but not in the iridophores. The current study nicely complements this work and identified a different gene (Tjp1a) which is acting only on the iridophores. However, the extent to which these two genetic pathways are connected needs to be resolved.

---

## [Author Response]

*1) Several previously published mutants (*luc, leo*) with similar pigment stripe defects disrupt connexins associated with gap junctions in pigment cells (though not specifically in iridophores), and in the Discussion the authors speculate that these may interact with Tjp1a. However, there are no experiments in the paper to address this hypothesis, such as tests of genetic interactions in* sbr;luc *and* sbr;leo *double mutants, evidence for physical interactions between proteins, or assays for functional gap junctions. Any evidence for (or against) the idea that loss of Tjp1a function disrupts gap junctions would greatly increase the paper's impact. The authors briefly mention that a reduction in Tjp1a may trigger iridophores to undergo the transition from dense to loose and more migratory behaviors. Would gap junctional communication likely influence this transition and if not what alternative types of cell-cell interactions might be involved?*

Tjp1a, which is known to interact with connexins in a specific fashion, is participating in the regulation of pattern formation as well. We included in the manuscript data from a yeast 2-hybrid experiment, which suggests that Tjp1a can interact with connexins39.4 and 41.8. However, both *luc* and *leo* connexins function in melanophores and xanthophores, but not in iridophores; this suggests that a direct interaction is unlikely, since Tjp1a is a cytoplasmic protein in iridophores. Hence we propose the existence of iridophore-specific membrane proteins, interacting with *luc* and *leo* connexins. This interaction would require Tjp1a in order to transmit a signal necessary for iridophore positioning and specification. As suggested, we included in the article data on double mutants. In both cases the phenotypes of double mutants are stronger than the ones of single mutants, suggesting that these genes are not forming a linear pathway in order to control pigmentation.

It was shown that many patterning mutants affect the capability of cells for homo- or heterotypic interactions and as a result their ability to aggregate (Inaba et al, Science, 2012; Maderspacher and Nüsslein-Volhard, Development, 2003; Eom et al, PLOS Genetics, 2013). Most of these studies investigated aggregation and dispersion behaviour of pigment cells. The mechanisms by which pigment cells mutually influence specification, differentiation and migration only recently became a focus of attention (Irion, *eLife*, 2014). Depolarization was suggested as a possible mechanism for interactions involving gap junctions (Inaba et al, Science, 2012). However, Tjp1a might also be involved in other as yet unknown mechanisms of cellular interactions.

*2) Much of the data is shown at low magnification and lacks cellular resolution: In*
Figure 2
*melanophores appear smaller in Tjp1a mutants, which could account for altered behaviors of iridophores. The authors argue instead that this is simply a difference in melanin distribution within melanophores of the same size, but this cannot be seen in the images provided. Mutant melanophores also appear smaller in*
Figure 4*.*

Thank you for pointing to us a possible misunderstanding. With the data presented in the Figure 5 we argue that melanin is concentrated in the centre of *sbr* melanophores, which appear as dots. This is due to the prolonged retention of their stellate, migratory shape, which can be seen in the Figure 5. However such a statement is indeed premature for Figure 2. This was corrected. The difference in shapes however, of course does not necessarily correlate with size, and we did not try to measure the sizes.

*Individual iridophores/xanthophores are also difficult to see in these figures*.

*Interpretations of iridophore behaviors depend on the ability to follow individual cells, such as TDL358:GFP+ cells in*
Figure 4*. The authors claim to have performed time-lapse imaging in the Results but do not appear to have tracked individual cells based on the still images provided and do not explain their time-lapsing procedures in the Methods. If not, some apparent cell movements may instead reflect changes in transgene expression or iridophore differentiation*.

The imaging shown in Figure 4 covers long times: about two weeks or longer (time varies from individual to individual, we use the size of the fish as a better measure of development). The figures were produced by imaging the individual fish once a day. Imaging sessions had to be very short (1-2 minutes) as the fish had to be completely paralyzed. Hence, we cannot provide more detailed images since it requires longer scanning times. We changed “time lapse” to “repeated imaging” in the text, as it was already referred to in the Materials and methods. Such imaging allows in some cases to follow individual melanophores, but due to the homogeneous appearance of iridophores, tracking of them is not possible. Naturally, we do not state that abnormal cell movement is the only possible cause of the iridophores invasion into dark stripe areas. Indeed, aberrant iridophore specification, namely compromised change from dense to loose form, is a more likely reason for that. We added more in-depth discussion on this subject to the manuscript.

*The supposed lack of immunostaining for Tjp1a in loose iridophores in*
Figure 7
*is unclear and difficult to distinguish from protein localized in surrounding epithelial cells*.

The image was changed to a more appropriate one.

*3) The information about the ultimate mechanism is a bit speculative. The phenotype should be corroborated using CRISPR technology*.

The phenotype of *sbr* mutants is clearly due to mutations in Tjp1a, we have five different alleles from two independent mutagenesis experiments. We agree that the exact mechanism by which pattern formation is affected in the mutants is not clear. However, given the fact that Tjp1a is a scaffolding protein that can interact with numerous other proteins we don’t have any clear candidates for CRISPR-mediated knock-outs.

*4) Previous work from the same group published in* eLife *(*[18]*) provided a genetic basis for a similar spotted phenotype. In this case the underlying connexin genes were required in the melanophores and xanthophores, but not in the iridophores. The current study nicely complements this work and identified a different gene (Tjp1a) which is acting only on the iridophores. However, the extent to which these two genetic pathways are connected needs to be resolved*.

We elaborated on this point in the text, using double mutants’ phenotypes and yeast 2-hybrid experiments.